# Social support and risk perception of influenza among Chengdu residents: A cross-sectional study during post-pandemic recovery

Cheng Yang[1,2], Jiawei Li[2,3], Qin Zeng[2,4], Xianqiong Feng[5]*

1 West China School of Nursing, Sichuan University/Department of Pediatric Intensive Care Unit Nursing, West China Second University Hospital, Sichuan University, Chengdu, China, 2 Key Laboratory of Birth Defects and Related Diseases of Women and Children (Sichuan University), Ministry of Education, Chengdu, China, 3 Department of Pediatric Outpatient Nursing, West China Second University Hospital, Sichuan University, Chengdu, China, 4 Department of Pediatrics Nursing, West China Second University Hospital, Sichuan University, Chengdu, China, 5 Department of Nursing, West China Hospital, Sichuan University/West China School of Nursing, Sichuan University, Chengdu, China

* fengxianqiong66@126.com

## Abstract

Influenza remains a global public health threat, yet limited evidence exists regarding public risk perception and social support dynamics in post-pandemic contexts. This study investigates the interplay between social support and influenza risk perception among residents in Chengdu, China, following the ease of COVID-19 restrictions. A cross-sectional survey was conducted from January to March 2024 using convenience sampling (n = 708). Validated tools included the Social Support Rating Scale (SSRS) and a researcher-developed Risk Perception Scale (RPS). Data were analyzed via Kruskal-Wallis tests and Spearman correlations using SAS 9.4 and SPSS 26.0. The results showed moderate median social support (37 points; 63.79% of maximum score) but alarmingly low risk perception (36 points; 40.0% of maximum score). Younger adults (<25 years) and older adults (>40 years) demonstrated significantly higher risk perception than middle-aged groups ($P < 0.01$). Healthcare workers exhibited substantially lower risk perception than students ($P < 0.01$). A weak but significant negative correlation emerged between social support and risk perception (r=−0.125, $P < 0.01$), with the strongest negative correlation observed for risk familiarity perception (r=−0.171, $P < 0.01$). Critically, Chengdu residents displayed inadequate risk awareness of influenza severity (score rate = 32%, very low) and familiarity (score rate = 36.7%, low), potentially undermining prevention efforts. Targeted health campaigns emphasizing risk communication (particularly for healthcare workers and middle-aged residents) and social resource mobilization are urgently needed to address these deficits.

**Data availability statement:** The data underlying the results presented in the study are available from the Zenodo repository [10.5281/zenodo.15389587], accessible via the link: https://zenodo.org/records/15389587

**Funding:** The author(s) received no specific funding for this work.

**Competing interests:** The authors have declared that no competing interests exist.

## Introduction

Influenza remains a significant global public health challenge, with an estimated 3−5 million severe cases and 290,000−650,000 deaths annually worldwide [1]. In China, influenza causes 84−144 million infections and up to 240,000 deaths each year [2,3]. Although the rigorous implementation of non-pharmaceutical interventions (NPIs) during the COVID-19 pandemic temporarily suppressed influenza transmission [4,5], a significant resurgence in activity intensity was observed after 2022. Sentinel hospitals in some southern Chinese provinces even reported influenza-like illness (ILI) percentages exceeding pre-pandemic levels [6,7]. indicating new challenges for public health.

Behavioral epidemiological studies indicate that public compliance with influenza prevention and control measures depends not only on epidemiological risks, but is also closely related to individuals' risk perception and social support – specifically the subjective assessment of disease threat severity (perceived severity), personal susceptibility (perceived susceptibility), and social resource support [8,9]. Therefore, the "rebound effect" in influenza prevention post-COVID-19 pandemic has revealed a critical research gap: in the post-pandemic era against the backdrop of relaxed non-pharmaceutical interventions (NPIs), what is the current status of public risk perception and social support, and how do they influence individuals to subsequently affect health behaviors for influenza prevention? However, current research based on China's influenza context remains very limited.

The Health Belief Model (HBM) provides a theoretical framework for this study. The model posits that health behavior decisions are driven by four-dimensional cognition: threat appraisal (severity, susceptibility), behavioral evaluation (benefit-barrier tradeoff), and external regulatory factors (cues to action, self-efficacy), which effectively guides the complex relationships among risk perception, social support, and health behaviors [10,11]. Recent empirical studies have found that social support may enhance the effect of HBM through dual pathways: on one hand, informational support (e.g., guidance from authoritative institutions) can serve as "cues to action" to directly trigger individual preventive behaviors; meanwhile, instrumental and emotional support (e.g., family assistance, psychological comfort) can indirectly promote behavioral change by reducing perceived barriers and enhancing self-efficacy [12–14].

Chengdu, as a megacity in southwestern China (with a permanent population exceeding 21 million), offers an ideal setting for this research. Firstly, Chengdu's 2023 influenza hospitalization rate was 37% higher than the national average, demonstrating typical post-pandemic epidemiological characteristics [15,16]; Secondly, during rapid urbanization, Chengdu exhibits significant occupational diversity, providing ample heterogeneous samples for this risk perception study [17,18]; Most importantly, Chengdu's unique collectivist cultural traditions and family support networks create a distinctive sociocultural laboratory for exploring how social support modulates risk perception (e.g., reducing threat severity appraisal and enhancing coping efficacy) through HBM pathways [19,20].

Based on the Health Belief Model (HBM), this study aims to analyze the public's influenza risk perception, social support status, and behavioral regulation mechanisms in post-pandemic Chengdu, so as to provide dual-dimensional "psychological-social" intervention strategies for influenza prevention and control, and to optimize the influenza prevention and control transmission framework for Chengdu and similar cities.

## Materials and methods

### Study design and participants

This was a cross-sectional survey design. To facilitate rapid data collection during the post-pandemic recovery phase, we employed a convenience sampling method across seven central urban districts with high population mobility (Jinjiang District, Wuhou District, Chenghua District, High-tech Zone, Qingyang District, Longquanyi District, and Shuangliu District) to capture diverse social demographics. Given schools' status as high-density environments and influenza hotspots, we expanded our study population to include adolescents with independent decision-making capacity, enabling comprehensive assessment of age-related differences in influenza risk perception. Inclusion criteria comprised: age 12–70 years; voluntary participation; Mandarin literacy. Exclusion criteria included: cognitive impairment, incomplete questionnaires (>20% missing data), or study withdrawal during data collection.

The study protocol received ethical approval from the Biomedical Ethics Review Board of Sichuan University (Approval No. WCSU-2022–787).The formal survey was administered from January 1 to March 31, 2024 via WJX platform (https://www.wjx.cn), a professional online survey tool widely adopted in China for its operational efficiency, data security, and anonymity features. Participants accessing the questionnaire were first presented with a detailed introductory page outlining study objectives, completion guidelines, and privacy protection measures. Platform settings restricted submissions to one response per IP address. Written informed consent was obtained from all participants prior to survey completion.

### Instruments

1. **Demographic Questionnaire:** Collected data on age, gender, occupation, income, and healthcare payment modalities.

2. **Social Support Rating Scale (SSRS):** This instrument employs a tri-dimensional framework (objective support, subjective support, and support utilization) comprising 10 standardized items. Scoring protocol: (1) Items 1–3 and 6–10 adopt a 4-point Likert scale (options A-D) with scores 1–4 reflecting incremental support levels from "none" to "full support"; (2) Pivotal items 4–5 utilize source enumeration: 0 points for no sources, with each documented source scoring 1 point. Total scores range 10–66, categorized via tertile segmentation: low (≤22), moderate (23–44), and high (≥45) support levels [21–23]. The scale demonstrated robust psychometric properties, having been validated across developmental stages (adolescents to adults) and diverse populations (healthcare workers, students, etc.) [24,25].

3. **Risk Perception Scale (RPS):** Developed through systematic literature review and theoretical framework construction [26–28], this scale addresses two critical limitations of existing instruments in Chinese cultural contexts: inadequate cultural specificity and absence of multidimensional risk constructs (e.g., core dimensions like risk familiarity and controllability). The finalized instrument contains four theoretical dimensions: (1) Risk familiarity (5 items) – assessing prior exposure awareness; (2) Risk controllability (5 items) – measuring perceived control capacity; (3) Risk severity (5 items) – quantifying consequence magnitude evaluation; (4) Risk susceptibility (3 items) – reflecting perceived exposure likelihood. All 18 items employ 5-point Likert scaling (1 = strongly disagree to 5 = strongly agree), with total scores 18–90 stratified into three risk perception tiers: low (18–42), moderate (43–66), and high (67–90).

Psychometric validation involved rigorous multistage testing: (1) Content validity assessment involving 24 public health and psychology experts demonstrated exceptional results, with item-level content validity indices (I-CVI)

ranging from 0.72 to 1.00 and scale-level validity (S-CVI/Ave) reaching 0.99, indicating thorough construct coverage and optimal item adequacy. (2) Reliability analysis revealed robust internal consistency (Cronbach's α = 0.822) and strong temporal stability evidenced by 2-week test-retest intraclass correlation coefficients (ICC = 0.85) in a substantial sample (N = 100). (3) Structural validation through prior exploratory analyses led to the removal of two items (f7, f8) with suboptimal factor loadings (<0.5), culminating in the finalized 18-item RPS questionnaire with enhanced psychometric properties [29,30].

## Statistical analysis

The statistical analysis was performed using SAS 9.4 and SPSS 26.0 software. Continuous variables that did not follow a normal distribution were expressed as median (first quartile, third quartile), and the Kruskal-Wallis rank-sum test was employed for univariate analysis. Categorical data were presented as frequency (percentage), with Spearman correlation analysis used to examine the relationships between variables. A *p*-value of <0.05 was considered statistically significant.

## Results and discussion

### Participant characteristics

We distributed 715 questionnaires and obtained 708 valid responses (98.9% response rate). The study population consisted of 602 female participants (85.0%), with a predominance of young adults: 465 respondents (65.7%) were aged <25 years (including 8 minors aged <18 years). Occupational distribution revealed that healthcare workers represented the largest subgroup (44.1%), followed by students (37.9%). A substantial proportion of participants (64.7%) reported monthly incomes ≤6000 CNY (Table 1).

**Table 1. Sociodemographic characteristics of participants.**

| Variable | n (%) |
|---|---|
| Age (years) | |
| <25 | 465 (65.68) |
| 25—30 | 78 (11.02) |
| 30—35 | 80 (11.30) |
| 35—40 | 39 (5.51) |
| ≥40 | 46 (6.49) |
| Sex | |
| Male | 106 (14.97) |
| Female | 602(85.03) |
| Occupation | |
| Healthcare worker | 312 (44.07) |
| Student | 268 (37.85) |
| Other* | 128 (18.08) |
| Monthly income (yuan) | |
| ≤6000 | 458 (64.69) |
| 6000~10000 | 145 (20.48) |
| 10000~20000 | 71 (10.03) |
| >20000 | 34 (4.80) |

Notes: Other*: include company employees, self-employed individuals, freelancers, teachers, etc.

### Risk perception scores

The median score of risk perception was 36 points, with a score rate of 40.00%. The median and score rates of each item were as follows: risk familiarity 11 points (36.67%), risk controllability 8 points (40.00%), risk severity 8 points (32.00%), and risk susceptibility 9 points (60.00%) (Table 2).

### Social support scores

The median score of social support was 37 points, and the score rate was 63.79%. The median scores and score rates of each item are as follows: Objective support 10 points (45.45%), subjective support 18 points (75.00%); The support utilization rate was 8 points (66.67%) (Table 3).

### Single-factor analysis of public risk perception

The influenza risk perception scores showed statistically significant variations across demographic strata (Table 4). Notably, two distinct age cohorts – participants under 25 years and those over 40 years – displayed elevated risk perception levels. Conversely, healthcare workers exhibited significantly lower risk perception compared to the general population ($p < 0.01$). Our analysis excluded subgroup comparisons for minors (n = 8) and seniors over 60 years (n = 2) due to insufficient sample sizes in these demographic groups. This limitation aligns with previous cross-linguistic studies demonstrating how sample size constraints can affect statistical power in demographic analyses.

### Correlation analysis

Total social support demonstrated a significant negative correlation with risk perception (r = −0.125, $p < 0.01$), with the strongest association observed for risk familiarity perception (r = −0.171, $p < 0.01$). Regarding positive correlations, objective support showed the most pronounced association with susceptibility (r = 0.100, $p < 0.05$) (Table 5 and Fig 1)

## Discussion

This study reveals persistently suboptimal influenza risk perception among Chengdu residents, particularly in disease severity assessment and pathogen familiarit – a pre-pandemic trend exacerbated post-COVID-19 [18,31]. The decline may reflect reduced vigilance after relaxation of non-pharmaceutical interventions (NPIs) [32], indicating a critical gap in

Table 2. Risk perception scores by dimension.

| Dimension | Total score | Median Score [Interquartile Range] | Score Rate (%) |
|---|---|---|---|
| Total | 18-90 | 36 (31-41) | 40.00 |
| Familiarity | 6-30 | 11 (6-12) | 36.67 |
| Controllability | 4-20 | 8 (8-10) | 40.00 |
| Severity | 5-25 | 8 (5-9) | 32.00 |
| Susceptibility | 3-15 | 9 (6-12) | 60.00 |

Table 3. Social support scores by dimension.

| Dimension | Total score | Median Score [Interquartile Range] | Score Rate (%) |
|---|---|---|---|
| Total | 558 | 37.00 (32.00,43.00) | 63.79 |
| Objective support | 122 | 10.00 (8.00,13.00) | 45.45 |
| Subjective support | 124 | 18.00 (15.00,23.00) | 75.00 |
| Utilization of support | 312 | 8.00 (7.00,10.00) | 66.67 |

**Table 4. Analysis of public risk perception scores for different sociological characteristics.**

| Variable | n (%) | Median Score [Interquartile Range] | Statistical value (H) | P value |
|---|---|---|---|---|
| Age (years) | | | 20.813 | <0.001 |
| <25 | 465 (65.68) | 37.00 (33.00,41.00) | | |
| 25—30 | 78 (11.02) | 34.00 (30.00,39.00) | | |
| 30—35 | 80 (11.30) | 35.00 (29.50,40.00) | | |
| 35—40 | 39 (5.51) | 33.00 (28.00,37.00) | | |
| ≥40 | 46 (6.49) | 37.50 (34.00,41.00) | | |
| Sex | | | 2.521 | 0.112 |
| Male | 106 (14.97) | 38.00 (31.00,41.00) | | |
| Female | 602(85.03) | 36.00 (31.00,41.00) | | |
| Occupation | | | 29.079 | <0.001 |
| Healthcare worker | 312 (44.07) | 34.00 (30.00,39.00) | | |
| Student | 268 (37.85) | 37.00 (33.00,42.00) | | |
| Other* | 128 (18.08) | 38.00 (33.00,42.00) | | |
| Monthly income (yuan) | | | 4.321 | 0.229 |
| ≤6000 | 458 (64.69) | 37.00 (32.00,41.00) | | |
| 6000-10000 | 145 (20.48) | 36.00 (31.00,40.00) | | |
| 10000-20000 | 71 (10.03) | 34.00 (29.00,40.00) | | |
| >20000 | 34 (4.80) | 35.00 (31.00,39.00) | | |

Notes: Other*: include company employees, self-employed individuals, freelancers, teachers, etc.

**Table 5. Spearman correlations between social support and risk perception (Spearman's r).**

| Dimension | Familiarity | Controllability | Severity | Susceptibility | Total score of Risk perception |
|---|---|---|---|---|---|
| Total score of support | −0.171^ | −0.126^ | −0.007* | 0.051* | −0.125^ |
| Objective support | −0.093^ | −0.055* | 0.017* | 0.100^ | −0.038* |
| Subjective support | −0.129^ | −0.088^ | −0.012* | −0.012* | −0.117^ |
| Support utilization | −0.164^ | −0.177^ | −0.041* | 0.046* | −0.125^ |

Notes: ^: $p < 0.05$; *: $p > 0.05$. Effect size: small (|0.10|–|0.29|), medium (|0.30|–|0.49|), large (≥|0.50|)

sustaining preventive behaviors post-crisis. Key demographic disparities emerged: both younger and older adults showed heightened risk awareness (likely due to age-related vulnerability perceptions), while healthcare workers exhibited desensitization from prolonged pathogen exposure. These findings necessitate tailored risk communication strategies addressing occupation-specific and age-related risk appraisal differences.

The study revealed moderate levels of social support overall, with subjective support components (e.g., familial care) demonstrating higher prevalence than objective support resources (e.g., institutional access). A statistically significant negative correlation emerged between total support and risk perception (r=−0.125, $p < 0.01$), consistent with the Health Belief Model's proposition that external support systems mitigate perceived prevention barriers. Notably, this inverse relationship was most pronounced for risk familiarity (r=−0.171, $p < 0.01$), indicating substantial cultural mediation effects. Within China's collectivist context, robust social networks appear to generate psychological buffering through three mechanisms: (1) Collective security attenuates individual vulnerability perceptions (e.g., anticipated reliance on others during illness); (2) High subjective support (scoring at the 75th percentile) strengthens relational safety networks; (3) Instrumental

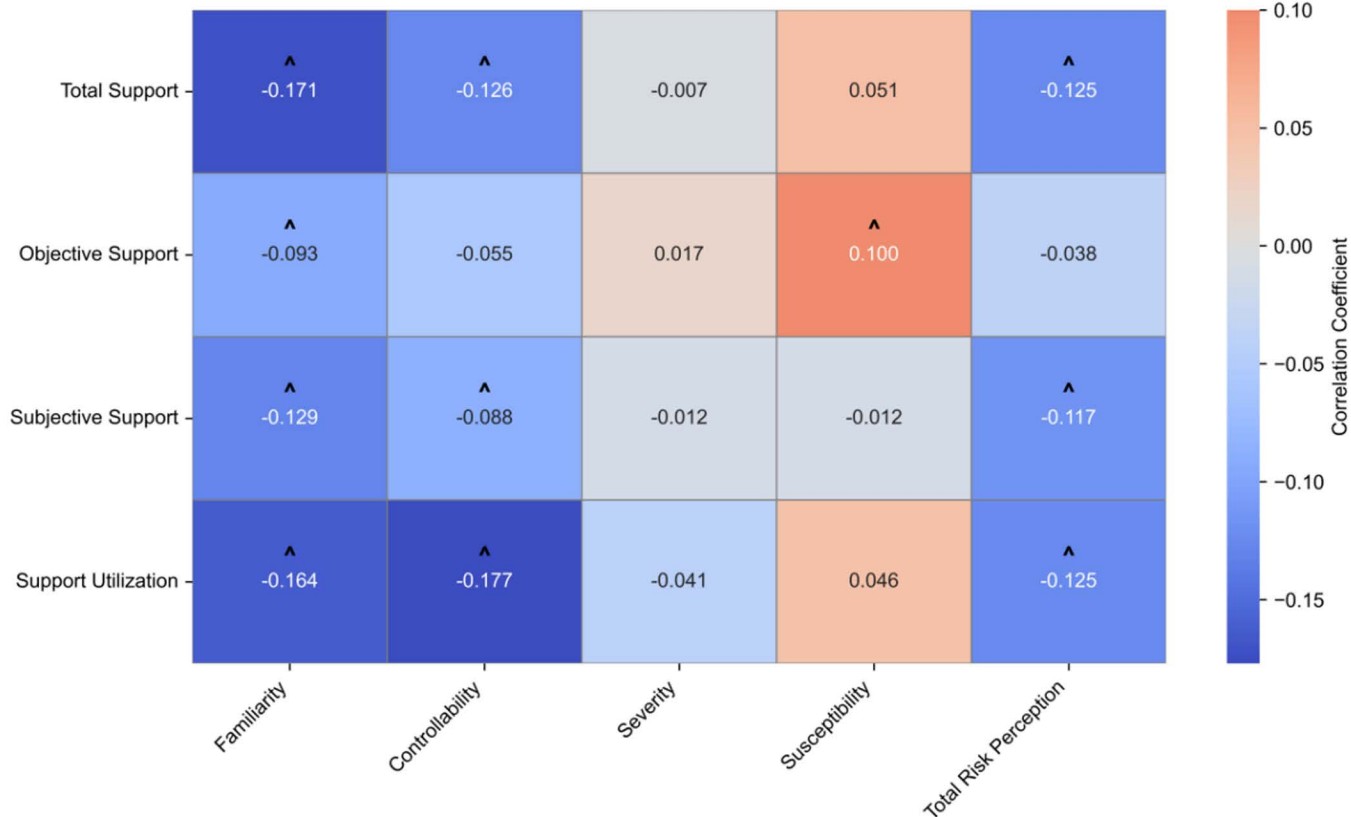

**Fig 1. The heatmap of social support and risk perception.**

aid reduces perceived severity. While social support theoretically functions as a behavioral cue, these findings suggest its primary role involves moderating threat appraisal through enhanced coping efficacy. However, the modest effect size (r < 0.30) implies limited direct behavioral influence, highlighting the predominant role of sociocultural norms (e.g., harmony-oriented collectivism) and structural interventions (e.g., policy-mediated resource distribution) in determining preventive health behaviors [33–35].

These results challenge health behavior models' overemphasis on social support. While subjective support may function as a Health Belief Model cue, its modest impact suggests greater emphasis should be placed on vaccination perceived benefits and sociocultural norms—particularly in contexts of institutional distrust or misinformation (unmeasured here). The subjective-objective support dissociation underscores the need for multilevel interventions combining community networks with structural healthcare access improvements.

The statistically significant but clinically marginal correlation highlights the necessity of distinguishing statistical outcomes from public health impact. Future research should employ longitudinal designs to analyze risk perception trends and mixed methods to examine cultural and media influences.

## Limitations

This investigation acknowledges three principal methodological constraints that warrant careful consideration:

1. Sampling bias: The utilization of convenience sampling—though operationally expedient for post-pandemic data collection—resulted in notable demographic disparities. The cohort exhibited substantial overrepresentation of females

(85.0%), young adults aged <25 years (65.7%), and healthcare workers (44.1%), thereby constraining the generalizability of our findings to the broader Chengdu population. Importantly, this sampling bias appears to have substantially influenced the observed risk perception patterns. Healthcare workers, comprising nearly half the sample, demonstrated significantly lower risk perception compared to students and other occupational groups ($p < 0.01$). This phenomenon may reflect occupational desensitization attributable to frequent pathogen exposure and specialized training, potentially leading to systematic underestimation of the population's true average risk perception level [36]. Conversely, the substantial student subgroup (37.9%), characterized by younger age and potentially heightened awareness stemming from recent pandemic experiences or educational environments, displayed elevated risk perception, possibly contributing to upward bias in the population mean estimate. Consequently, the marked occupational imbalance (predominance of healthcare and student populations) suggests that our reported overall risk perception score (median = 36, score rate = 40.0%) may not accurately represent the true distribution across diverse demographic strata in Chengdu. Future investigations employing probability sampling designs are warranted to validate these patterns and precisely quantify risk perception among currently underrepresented populations, including older adults, manual laborers, and rural-to-urban migrants [37].

2. Measurement granularity: Analytical precision was constrained by insufficient occupational classification resolution. Diverse professional groups (e.g., educators, independent contractors) were aggregated into a single "Other" category (18.0%), precluding nuanced cross-occupational comparisons of risk perception paradigms. Concurrently, critical developmental stages – particularly adolescents (1%) and older adults (0.3%) – were substantially underrepresented, potentially obscuring lifespan-specific vulnerability trajectories.

3. Generalizability constraints: Despite intentional inclusion of varied income strata and an age spectrum spanning 12–70 years, the sampling framework failed to adequately capture non-institutionalized populations (e.g., rural dwellers, informal sector workers). This omission may limit the external validity of findings when extrapolated to broader urban contexts beyond the study's specific institutional catchment areas.

These limitations collectively underscore the need for cautious interpretation of results and highlight critical directions for future research employing more representative sampling strategies.

## Future directions

To address these limitations, future studies should implement: 1) Stratified random sampling to capture underrepresented populations (e.g., elderly, adolescents, and non-healthcare workers); 2) Multi-channel recruitment strategies (e.g., community-based sampling) to improve population representativeness; 3) Purposive sampling of occupational subgroups for cross-professional comparisons; 4) Longitudinal mixed-methods designs to examine unmeasured confounders like cultural narratives and structural inequities in risk perception dynamics.

## Conclusion

Our study reveals consistently low influenza risk perception among Chengdu residents during the post-COVID-19 period, particularly regarding perceived severity and disease familiarity. Key demographic disparities emerged across age groups and occupations. While social support showed moderate association with risk perception, socio-cultural norms and structural factors exerted stronger influences on preventive behaviors. These findings underscore the need for:

• Tailored risk communication interventions addressing occupation-specific vulnerabilities

• Equitable public health resource allocation (e.g., vaccine subsidy programs)

• Representative longitudinal studies to track post-pandemic risk perception evolution

## Acknowledgments

We sincerely appreciate all volunteers for their invaluable contributions to this study. Special thanks to our research team for their expertise and dedication throughout this project.

## Author contributions

**Data curation:** Cheng Yang.

**Formal analysis:** Jiawei Li.

**Investigation:** Jiawei Li.

**Methodology:** Xianqiong Feng.

**Project administration:** Xianqiong Feng.

**Supervision:** Xianqiong Feng.

**Writing – original draft:** Cheng Yang, Qin Zeng.

**Writing – review & editing:** Cheng Yang.

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
