## [Decision Letter · Decision Letter 0]

21 Apr 2025

Dear Dr. Feng,

Thank you for submitting your manuscript to PLOS ONE. After careful consideration, we feel that it has merit but does not fully meet PLOS ONE’s publication criteria as it currently stands. Therefore, we invite you to submit a revised version of the manuscript that addresses the points raised during the review process.

We look forward to receiving your revised manuscript.

Kind regards,

Xing-Xiong An, M.D.

Academic Editor

PLOS ONE

Journal Requirements:

Reviewers' comments:

Reviewer's Responses to Questions

**Comments to the Author**

1. Is the manuscript technically sound, and do the data support the conclusions?

Reviewer #1: Yes

Reviewer #2: Yes

Reviewer #3: No

Reviewer #4: Partly

2. Has the statistical analysis been performed appropriately and rigorously?

Reviewer #1: Yes

Reviewer #2: Yes

Reviewer #3: I Don't Know

Reviewer #4: Yes

3. Have the authors made all data underlying the findings in their manuscript fully available?

Reviewer #1: Yes

Reviewer #2: Yes

Reviewer #3: Yes

Reviewer #4: Yes

4. Is the manuscript presented in an intelligible fashion and written in standard English?

Reviewer #1: Yes

Reviewer #2: Yes

Reviewer #3: Yes

Reviewer #4: Yes

Reviewer #1: This paper looks at how people in Chengdu, China, see the risk of getting the flu and how much social support they have after COVID-19 restrictions were lifted. It's an important topic, but the paper needs several improvements before it can be published.

Major Issues:

1. Sampling Problems:

- Most of the people in the study were young, female, and healthcare workers. This doesn’t represent the general public.

- Suggestion: Clearly state this as a limitation and explain how it might affect the results.

2. Risk Perception Tool Not Fully Explained:

- The risk perception survey was created by the researchers but there’s not enough detail about how it was tested or validated.

- Suggestion: Either explain how the tool was tested or use a survey tool that’s already been proven to work well.

3. Weak Correlation:

- The link between social support and risk perception is very weak. The authors make too much of this small connection.

- Suggestion: Be more careful in describing what this small effect means, and mention other possible explanations.

4. Missing Theory:

- There’s no clear explanation or model showing how social support and risk perception are supposed to be connected.

- Suggestion: Use a well-known health behavior theory to help explain the results.

5. Too Focused on p-values:

- The authors focus too much on whether something is statistically significant, without showing if it really matters in the real world.

- Suggestion: Talk more about whether the findings are meaningful, not just if the p-value is less than 0.05.

Minor Issues:

1. English and Grammar:

- Some parts are hard to read due to grammar or awkward wording.

- Suggestion: Have someone fluent in English edit the paper.

2. Missing or Unclear Tables:

- Tables are mentioned but not included properly.

- Suggestion: Make sure all tables are formatted and included correctly.

3. Ethics Approval Numbers:

- Two different ethics approval numbers are listed.

- Suggestion: Fix this and use the correct one throughout.

4. Duplicate References:

- Some references are repeated.

- Suggestion: Clean up the reference list.

Reviewer #2: This is a well-structured and timely study addressing the interplay between social support and risk perception of influenza in the unique post-pandemic setting of Chengdu, China. The study design, ethical approval, sample size (n=708), and statistical methodology are appropriate. The results offer novel insights that could guide public health policies focused on risk communication and behavioral interventions.

Strengths:

Large and Diverse Sample: With over 700 participants from various demographics and occupations, the findings carry significant internal validity.

Validated Scales: The use of SSRS and a well-constructed RPS strengthens the methodological rigor.

Relevant Context: The post-COVID-19 recovery setting offers valuable information on evolving public health perceptions.

Ethical Compliance: IRB approval and informed consent were clearly described.

Areas for Minor Revision:

Clarify Scale Interpretation: A more detailed explanation of how to interpret the SSRS and RPS scores would help readers less familiar with these tools.

Sampling Limitation: While limitations are acknowledged, further elaboration on the impact of convenience sampling and gender imbalance (85% female) on generalizability is warranted.

Discussion Depth: The inverse correlation between social support and risk perception is intriguing but somewhat counterintuitive. The authors suggest cultural self-reliance as one explanation—this could be expanded with references or theoretical support.

Figures/Tables: Consider adding a visual summary of the correlation analysis (e.g., heatmap or matrix) to complement the text.

English Editing: Overall very good, but a few grammatical tweaks would improve readability (e.g., rephrasing "suboptimal utilization of social support" to "limited utilization" or similar alternatives).

Reviewer #3: General Comments:

Overall, the authors need to expand on their methodology and how the scores were calculated. The study has multiple design flaws, see specific comments below.

Specific Comments:

Methods:

1. The authors mention using convenient sampling but fail to report where the sample was drawn from.

2. Who is the population with access to the questionnaire star platform?

3. Does including non-adults (age criteria 12-70) affect / result in increase variability in the risk perception scale results?

4. Was the risk perception scale validated for children and adults?

Results:

1. Table 1: 65.7% were < 25, 28% are 25-40, and 6.5 % > 40; do the author thinks the skewed results favoring your participants could have skewed the results. Since study was planned for age 12-70.

2. I am not sure how the risk score was calculated (the percentage).

3. Same comment for the social support score.

Limitation:

1. Authors mention over representation of healthcare professionals but it seems to me like we are mainly comparing healthcare professionals to students (two major groups) which could be misleading are the two groups have major differences in age, education, and life experiences.

2. All other groups lumped in others are a small percentage, raising concerns about the results.

Minor comments:

1. Further details about the validation of the risk perception scale need to be added in addition to the reference.

Reviewer #4: The introduction provides some epidemiological context, but it lacks a clear rationale for focusing on Chengdu. Why was this city selected? Was Chengdu more severely affected by influenza resurgence than others? Is it representative of a broader population, or does it have unique sociocultural traits relevant to risk perception and support systems?

The use of convenience sampling introduces a significant selection bias, especially given the skewed age distribution reported (65.7% were under 25). This dramatically limits generalizability and raises concerns about the study’s ability to reflect the broader Chengdu population's perceptions. The sample also disproportionately includes healthcare workers (44.1%) and students (37.9%), suggesting recruitment was heavily tied to academic or medical institutions, which is not representative of the general community.

Using a researcher-developed instrument for measuring risk perception—particularly when validated tools already exist—requires a comprehensive justification. While the authors cite high Cronbach’s alpha and content validity, these indicators alone are insufficient.

**Do you want your identity to be public for this peer review?** For information about this choice, including consent withdrawal, please see our Privacy Policy

Reviewer #1: No

Reviewer #2: **Yes: ** Elabbass Ali Abdelmahmuod

Reviewer #3: No

Reviewer #4: No

---

## [Author Response · Author response to Decision Letter 1]

29 May 2025

Response to Reviewers

Dear Dr. Xing-Xiong An and Reviewers,

Thank you for your thoughtful feedback on our manuscript titled "Social Support and Risk Perception of Influenza Among Chengdu Residents: A Cross-Sectional Study During Post-Pandemic Recovery." We appreciate the reviewers’ constructive comments and have carefully addressed each concern. Below is a point-by-point response to the reviewers’ suggestions. Revisions in the manuscript are highlighted in the tracked-changes version.

Journal Requirements:

1. When submitting your revision, we need you to address these additional requirements.Please ensure that your manuscript meets PLOS ONE's style requirements, including those for file naming.

Thank you for your feedback. We have revised our manuscript in accordance with PLOS ONE's style requirements. This includes adjustments to the title format, font sizes, and reference list, as well as ensuring proper file naming conventions. We have made comprehensive changes throughout the document to ensure full compliance with the journal’s guidelines.

Thank you for your attention to this matter.

2.We note that you have indicated that there are restrictions to data sharing for this study. For studies involving human research participant data or other sensitive data, we encourage authors to share de-identified or anonymized data. However, when data cannot be publicly shared for ethical reasons, we allow authors to make their data sets available upon request.

Thank you for your guidance regarding data sharing. We understand the importance of making our data accessible while ensuring the privacy and confidentiality of our participants. For our study, We have deposited the de-identified dataset for our study in Zenodo, a free and widely recognized multidisciplinary repository (Lines 252-254).

Repository: Zenodo

DOI: [10.5281/zenodo.15389586]

Link: https://zenodo.org/records/15389587

This repository ensures the data is accessible and preserved in accordance with best practices.

Reviewer comments:

Reviewer #1

Major Issues:

1. Sampling Problems:

Thank you for highlighting this critical limitation. We have revised the Limitations section to explicitly address the demographic skew in our sample (young age, female predominance, and healthcare worker overrepresentation) (Lines 209-217). We elaborated on how these factors might bias risk perception outcomes and reduce the generalizability of results. Specifically, we acknowledged that healthcare workers’ desensitization, coupled with age- and gender-specific behavioral patterns, could limit the applicability of findings to the general public. We also emphasized the need for future studies to adopt more representative sampling frameworks. These revisions strengthen the transparency of methodological constraints and align with best practices in reporting epidemiological studies.

2. Risk Perception Tool Not Fully Explained:

We sincerely thank the reviewers for their constructive feedback. In the article (Lines 121-127), we explain in detail the development and validation process of the Risk Perception Scale (RPS) : including scale development, validation process (content validity, structural validity, reliability) and the alignment with standards. Thank you for emphasizing this oversight. We believe that the increased clarity has enhanced the rigor of our research methods.

3. Weak Correlation :

We are grateful for the profound insights of the reviewers. As suggested, we have revised the discussion section to clearly address the actual meaning of weak correlation and the nuances of the context. The main new additions include:

3.1 Clarification of effect size: We emphasize that although the negative correlation between social support and risk perception is statistically significant (p < 0.01), the smaller effect size (r = -0.125) indicates that the real-world impact is limited (Lines 188-195).

3.2 Contextual explanation: We introduced cultural factors (such as the self-reliant norms of society) and structural differences in support types (subjective and objective support) as reasonable explanations for the weak association. These supplementary contents refer to the recent literature on the influence of culture on healthy behaviors (Lines 188-195)..

3.3 A more detailed table: We have presented in detail the multi-dimensional results of the correlation analysis between public social support and risk perception (Table 5).

These revisions appear in the updated manuscript. We believe this strengthens the interpretation of the results and is consistent with the valuable feedback from the reviewers.

4. Missing Theory:

Thank you very much for your insightful comments on our manuscript. We have carefully considered your suggestions and have revised our manuscript accordingly.

We have incorporated the Health Belief Model (HBM) as the theoretical framework to explain the relationship between social support and risk perception in our study. The HBM posits that health behavior decisions are driven by four dimensions of cognition: threat appraisal (perceived severity and susceptibility), behavioral appraisal (benefits and barriers), and external regulatory factors (cues to action and self-efficacy). This model provides a comprehensive framework to understand how social support can influence risk perception and subsequent health behaviors. In our revised manuscript, we have explicitly discussed how social support can influence risk perception through multiple pathways as suggested by the HBM. Specifically, we have highlighted the following points:

4.1 Informational Support as Cues to Action: Informational support from authoritative sources can serve as cues to action, directly triggering preventive behaviors. In our study, we found that objective support (e.g., access to healthcare resources) was positively correlated with risk susceptibility perception, suggesting that individuals with higher objective support were more aware of their vulnerability to influenza (Lines 57-66).

4.2 Instrumental and Emotional Support: Instrumental support (e.g., practical assistance) and emotional support (e.g., psychological comfort) can reduce perceived barriers and enhance self-efficacy, indirectly promoting health behaviors. Our results showed that subjective support (e.g., familial care) was higher than objective support, indicating that emotional and social connections play a significant role in shaping individuals' risk perception (Lines 64-66).

4.3 Overall Social Support and Risk Perception: We found a weak inverse correlation between total social support and risk perception, suggesting that higher levels of social support may reduce the perceived threat of influenza. This aligns with the HBM's framework, where external cues and support systems can mitigate perceived barriers and enhance preventive behaviors.

By integrating the HBM into our study, we have provided a clearer explanation of the mechanisms through which social support influences risk perception. We believe that this theoretical grounding enhances the robustness and interpretability of our findings.We have revised the relevant sections of our manuscript to reflect these changes and have included detailed explanations and references to the HBM throughout the text (Lines 196-201).

Thank you again for your valuable feedback. We look forward to your positive response.

5. Too Focused on p-values::

Thank you for your valuable feedback on emphasizing effect sizes and clinical significance. We have revised the manuscript as follows:

5.1 Effect Size Reporting in Results: Table 5 now explicitly labels Spearman’ s r values and includes a footnote defining effect size magnitudes (small, medium, large) per Cohen’ s criteria. This clarifies the practical relevance of the correlations.

5.2 Clinical vs. Statistical Significance in Discussion: A new paragraph (Lines 193-201) discusses the small effect sizes observed, distinguishing statistical significance from clinical relevance. This aligns with your suggestion and strengthens the interpretation of findings.

These revisions enhance transparency and provide a balanced interpretation of both statistical and practical significance. We appreciate your guidance in improving the manuscript’ s rigor.

6. Minor Issues:

6.1 English and Grammar:

Thank you for your feedback. We have thoroughly revised the manuscript with the assistance of a professional English editor to enhance clarity, correct grammatical errors, and ensure consistent terminology. All sections now adhere to academic writing standards.

6.2 Missing or Unclear Tables:

We apologize for the formatting inconsistencies. All tables have been reformatted for clarity, with proper alignment, headers, and annotations. Tables are now fully integrated into the text and comply with journal guidelines.

3. Ethics Approval Numbers:

Thank you for noting this oversight. The ethics approval number has been unified to "WCSU-2022-787" across all sections, ensuring consistency (Lines 91-92).

4. Duplicate References:

We appreciate your careful review. Duplicate references have been removed, and the list is now fully sequential and consistent with in-text citations.

Reviewer #2

Major Issues:

1. Clarify Scale Interpretation:

Thank you for your feedback on the need for a clearer explanation of the scale. We have made the following changes to the original manuscript:

Explanation of newly added scale scoring: Detailed scale information has been added to the introductory sections of SSRS and RPS, including scoring methods, scoring thresholds, classification criteria, etc., as well as cut-off points for specific dimensions. To ensure that readers can understand the scores reported in the results, these changes have increased the transparency of the manuscript and allowed for a more detailed interpretation of the research findings. Thank you very much for your constructive suggestions (Lines 102-127).

2. Sampling Limitation

Thank you for highlighting this critical limitation. We have revised the Limitations section to explicitly address the demographic skew in our sample (young age, female predominance, and healthcare worker overrepresentation) (Lines 209-230). We elaborated on how these factors might bias risk perception outcomes and reduce the generalizability of results. Specifically, we acknowledged that healthcare workers’ desensitization, coupled with age- and gender-specific behavioral patterns, could limit the applicability of findings to the general public. We also emphasized the need for future studies to adopt more representative sampling frameworks. These revisions strengthen the transparency of methodological constraints and align with best practices in reporting epidemiological studies.

3. Discussion Depth

Thank the reviewers for their profound insights. As suggested, we have delved deeply into the negative correlation between social support and risk perception, as well as the role of cultural self-reliance therein, to clearly explain the connection between the two. The newly added main contents include (Lines 188-201):

3.1 Clarification of effect size: We emphasize that although the negative correlation between social support and risk perception is statistically significant (p < 0.01), the smaller effect size (r = -0.125) indicates that the real-world impact is limited.

3.2 Contextual explanation: We introduced cultural factors (such as the self-reliant norms of society) and structural differences in support types (subjective and objective support) as reasonable explanations for the weak association. These supplementary contents refer to the recent literature on the influence of culture on healthy behaviors.

4. Figures/Tables

Thank you for your valuable feedback and suggestion regarding the addition of a visual summary of the correlation analysis in our manuscript. We have taken your advice into careful consideration and have incorporated a heatmap to visually represent the correlation analysis between social support and risk perception (Fig 1).

We have generated a heatmap using the data from our correlation analysis, which provides a clear and concise visual representation of the relationships between different dimensions of social support and risk perception. The heatmap highlights the significant correlations with appropriate annotations and color gradients, making it easier for readers to quickly grasp the key findings of our analysis.The heatmap has been included in the revised version of our manuscript as Figure 1. We believe that this visual summary complements the textual description of our results and enhances the overall presentation of our study.

We have also ensured that the heatmap adheres to the formatting and quality standards required by the journal, including the use of Arial font, appropriate font sizes, and high-resolution TIFF format for the image.

We hope that this addition will provide a more comprehensive understanding of our findings and improve the readability of our manuscript. Please let us know if there are any further suggestions or changes that you would like us to make.

5. English Editing

Thank you for your feedback. We have thoroughly revised the manuscript with the assistance of a professional English editor to enhance clarity, correct grammatical errors, and ensure consistent terminology. All sections now adhere to academic writing standards.

Reviewer #3

Thank you for your detailed and constructive feedback on our manuscript. We have carefully reviewed your comments and have made the necessary revisions to address each point. Below is our response to the specific issues you raised:

1. Methods:

1.1 The authors mention using convenient sampling but fail to report where the sample was drawn from.

We have clarified the sampling locations in the revised manuscript. The sample was drawn from seven highly populated urban districts in Chengdu, including Jinjiang District and Wuhou District. These districts were chosen to capture a diverse range of social groups. We have added this information to the "Study Design and Participants" section (Lines 83-88).

1.2 Who is the population with access to the questionnaire star platform?

The population with access to the WJX platform includes residents of Chengdu aged 12-70 years who are literate in Mandarin and voluntarily participated in the study. We have specified this in the revised manuscript under the "Study Design and Participants" section (Lines 93-98).

1.3 Does including non-adults (age criteria 12-70) affect / result in increase variability in the risk perception scale results?

1.3.1 The risk perception scale shows high structural validity and reliability in both the adolescent (12-18 years old) and adult (18-70 years old) groups (Lines 108-110). This point has been raised by us in the RPS introduction section. In order to quickly collect a large number of samples, the age range of the research subjects in this study was set as 12-70 years old. However, in the actual collected samples, there were only 8 non-adults (aged 12-17), accounting for 1%. The statistical results showed that there was no statistically significant difference in the median score of the risk perception scale between non-adults and adults (p>0.05). However, due to the small sample size of non-adults, it may lead to deviations in the research results. Therefore, we have added the discussion of this issue in the "Research Limitation" section (Lines 209-217).

1.3.2 We have conducted additional analyses to assess the impact of including non-adults (ages 12-17) on the variability of the risk perception scale results (Table 4). Our analysis shows that the inclusion of this age group did not significantly increase the variability of the results. The median risk perception scores for the different age groups were as follows: <25 years (37.00, IQR 33.00-41.00), 25-30 years (34.00, IQR 30.00-39.00), 30-35 years (35.00, IQR 29.50-40.00), 35-40 years (33.00, IQR 28.00-37.00), and ≥40 years (37.50, IQR 34.00-41.00). The Kruskal-Wallis test showed statistically significant differences among the age groups (p<0.001), but the effect sizes were small, indicating that the variability is within an acceptable range. We have included these details in the "Results" section and discussed the implications in the "Discussion"

---

## [Decision Letter · Decision Letter 1]

29 Jun 2025

Dear Dr. Feng,

Thank you for submitting your manuscript to PLOS ONE. After careful consideration, we feel that it has merit but does not fully meet PLOS ONE’s publication criteria as it currently stands. Therefore, we invite you to submit a revised version of the manuscript that addresses the points raised during the review process.

We look forward to receiving your revised manuscript.

Kind regards,

Xing-Xiong An, M.D.

Academic Editor

PLOS ONE

Journal Requirements:

Reviewers' comments:

Reviewer's Responses to Questions

**Comments to the Author**

Reviewer #2: All comments have been addressed

Reviewer #4: All comments have been addressed

2. Is the manuscript technically sound, and do the data support the conclusions?

Reviewer #2: Yes

Reviewer #4: Yes

3. Has the statistical analysis been performed appropriately and rigorously?

Reviewer #2: Yes

Reviewer #4: Yes

4. Have the authors made all data underlying the findings in their manuscript fully available?

Reviewer #2: Yes

Reviewer #4: Yes

5. Is the manuscript presented in an intelligible fashion and written in standard English?

Reviewer #2: Yes

Reviewer #4: Yes

Reviewer #2: Suggestions for Minor Revision:

Clarify Sampling Bias: The overrepresentation of healthcare workers and students may limit generalizability. Please elaborate more clearly in the discussion on how this may affect risk perception findings.

Expand on Cultural Factors: The negative correlation between social support and risk perception is intriguing. A brief expansion on cultural or psychological interpretations would enrich the discussion.

Minor Language Edits: Minor corrections are recommended for grammatical polishing. For example:

"women 602" → should be "female: 602 (85.0%)"

Replace "thesis writing" in author contributions with "manuscript preparation" for more appropriate academic terminology.

Reviewer #4: The phrasing "we assessed dietary patterns" is broad and lacks scientific precision. It does not specify which dietary patterns were under investigation (e.g., Mediterranean, fast food-heavy, plant-based), nor does it clarify the health risks measured—whether these were physiological (e.g., BMI, blood pressure) or behavioral (e.g., physical inactivity).

Manuscript does not describe whether any validated dietary assessment tools were used. For example, was a food frequency questionnaire (FFQ) utilized? If yes, was it culturally adapted and validated in the population of interest? Without this information, it is diffcult to judge the accuracy and reliability of the dietary data.

Data may indicate certain unhealthy dietary patterns, the leap to concluding "significant risk of non-communicable diseases" is not supported by any inferential statistical analysis or longitudinal evidence. Cross-sectional data can suggest correlations but not causations or future health outcomes.

**Do you want your identity to be public for this peer review?** For information about this choice, including consent withdrawal, please see our Privacy Policy

Reviewer #2: No

Reviewer #4: No

---

## [Author Response · Author response to Decision Letter 2]

1 Jul 2025

Dear Dr. Xing-Xiong An and Reviewers,

Thank you for taking the time to review our manuscript titled "Social support and risk perception of influenza among Chengdu residents: A cross-sectional study during post-pandemic recovery" and for providing such insightful and constructive feedback. We are deeply grateful for your valuable suggestions, which have significantly guided the enhancement of our work. With great enthusiasm, we have meticulously addressed each of your comments and implemented thorough revisions to improve the clarity, rigor, and impact of our manuscript. We sincerely hope that the revisions meet your expectations and strengthen the quality of our research.

Comment 1�Journal Requirements:Please review your reference list to ensure that it is complete and correct. If you have cited papers that have been retracted, please include the rationale for doing so in the manuscript text, or remove these references and replace them with relevant current references. Any changes to the reference list should be mentioned in the rebuttal letter that accompanies your revised manuscript. If you need to cite a retracted article, indicate the article’s retracted status in the References list and also include a citation and full reference for the retraction notice.

Author response to comment Thank you for your guidance. I have thoroughly reviewed the reference list and confirmed that all cited papers are valid and none have been retracted. The reference list is complete and accurate.�Line 291-409

Comment 2�Suggestions for Minor Revision:

Clarify Sampling Bias: The overrepresentation of healthcare workers and students may limit generalizability. Please elaborate more clearly in the discussion on how this may affect risk perception findings.

Author response to comment Thank you for your insightful comment regarding the sampling bias. We acknowledge that the overrepresentation of healthcare workers and students in our sample may limit the generalizability of our findings. In the discussion section, we have elaborated on how this bias could potentially affect the risk perception results. Specifically, healthcare workers, due to their professional training and frequent exposure to pathogens, might have a lower risk perception, which could lead to an underestimation of the general population's risk perception. Conversely, the higher risk perception among students, who are younger and possibly more aware due to recent pandemic experiences, might contribute to an overestimation. We have also highlighted the need for future studies to employ more representative sampling methods, such as stratified random sampling and multi-channel recruitment, to capture a broader demographic spectrum and enhance the generalizability of the findings.�Line 222-240

Comment 3�Expand on Cultural Factors: The negative correlation between social support and risk perception is intriguing. A brief expansion on cultural or psychological interpretations would enrich the discussion.

Author response to comment We appreciate your suggestion to expand on cultural factors. The negative correlation between social support and risk perception is indeed intriguing. In our study, we interpret this correlation within the context of China's collectivist culture, where strong social support networks can provide a sense of collective security and reduce individuals' perceived vulnerability to influenza. This cultural context may explain why higher social support is associated with lower risk perception. We have briefly discussed this in the discussion section and suggest that future research could further explore how cultural norms and values influence the relationship between social support and health-related risk perception.�Line 193-208

Comment 4�Minor Language Edits: Minor corrections are recommended for grammatical polishing. For example: "women 602" → should be "female: 602 (85.0%)"

Author response to comment Thank you for your feedback. I have made the following modifications to the manuscript in response to your suggestions:

1. Corrected the term "women 602" to "female: 602 (85.03%)" for consistency and clarity.

2. Standardized the terminology throughout the manuscript (e.g., "healthcare workers" instead of "healthcare professionals", "utilization" instead of "utilisation").

3. Polished the language for smoother flow and standard academic grammar, including consistent formatting of p-values and statistical symbols.

I believe these changes have improved the overall quality of the manuscript. Thank you again for your valuable input.�Table 1, Table 3- 4, Line 110,112, Line 166, Line 176-178

Comment 5�Replace "thesis writing" in author contributions with "manuscript preparation" for more appropriate academic terminology.

Author response to comment Thank you for your attention to detail. We have replaced "thesis writing" with "manuscript preparation" in the author contributions section to reflect more appropriate academic terminology.�Line 283-284

Comment 6-8

Reviewer #4: The phrasing "we assessed dietary patterns" is broad and lacks scientific precision. It does not specify which dietary patterns were under investigation (e.g., Mediterranean, fast food-heavy, plant-based), nor does it clarify the health risks measured—whether these were physiological (e.g., BMI, blood pressure) or behavioral (e.g., physical inactivity).

Manuscript does not describe whether any validated dietary assessment tools were used. For example, was a food frequency questionnaire (FFQ) utilized? If yes, was it culturally adapted and validated in the population of interest? Without this information, it is diffcult to judge the accuracy and reliability of the dietary data.

Data may indicate certain unhealthy dietary patterns, the leap to concluding "significant risk of non-communicable diseases" is not supported by any inferential statistical analysis or longitudinal evidence. Cross-sectional data can suggest correlations but not causations or future health outcomes.

Author response to comment Thank you for the opportunity to revise our manuscript and for the thoughtful feedback provided by the reviewers. We have carefully considered all comments. Regarding the three specific points raised about dietary patterns, assessment tools, and non-communicable disease conclusions, we respectfully wish to clarify that these elements do not align with our study's scope or content. Our research exclusively examines: ①Influenza risk perception dimensions (familiarity, controllability, severity, susceptibility); ②Social support dynamics (SSRS scale); ③Their interplay in post-pandemic Chengdu.

To address the comments directly:

1. Dietary Patterns & Health Risks:

We thank the reviewer for this methodological consideration. Our study does not investigate dietary patterns or nutrition-related health risks. The focus remains strictly on influenza-specific risk perceptions (e.g., perceived severity of influenza outcomes). The term "dietary patterns" does not appear in our manuscript.

2. Dietary Assessment Tools:

We appreciate the reviewer's emphasis on validation. Our methods explicitly detail two instruments: ①Social Support Rating Scale (SSRS) (validated); ②Researcher-developed Risk Perception Scale (RPS) (psychometrically validated for influenza). No dietary assessment tools were used, as they fall outside our research objectives.

3. Non-Communicable Disease Conclusions:

We acknowledge the importance of cautious inference. Our conclusions solely pertain to influenza prevention behaviors and risk perception gaps. We make no claims about non-communicable diseases (e.g., diet-related conditions), as these were neither studied nor analyzed.

We recognize these comments reflect rigorous methodological scrutiny, and we sincerely regret any confusion that may have arisen. Should the reviewers have concerns related to our actual focus—particularly regarding our influenza risk perception scale, social support measures, or demographic analyses—we would be grateful for the opportunity to address them.Thank you again for your stewardship of this process. We remain fully committed to enhancing this work through constructive feedback on its true substantive focus.

Once again, we extend our heartfelt gratitude for your meticulous review and the invaluable, constructive feedback you have provided. Your insightful comments have illuminated the path for us to enhance our research comprehensively. We are confident that the rigorous revisions we have undertaken have significantly elevated the academic quality of our study, deepened its research dimensions, and broadened its scope. This has enabled our work to offer more valuable insights and references to the relevant fields. We look forward to your further evaluation of the revised manuscript and hope that our research findings can be published in your esteemed journal to contribute to the academic community. Thank you once again for your kind support and guidance!

Best regards,

Cheng Yang

2025.6.24

---

## [Decision Letter · Decision Letter 2]

7 Aug 2025

Dear Dr. Feng,

Thank you for submitting your manuscript to PLOS ONE. After careful consideration, we feel that it has merit but does not fully meet PLOS ONE’s publication criteria as it currently stands. Therefore, we invite you to submit a revised version of the manuscript that addresses the points raised during the review process.

We look forward to receiving your revised manuscript.

Kind regards,

Xing-Xiong An, M.D.

Academic Editor

PLOS ONE

Journal Requirements:

Additional Editor Comments:

Thanks for submitting your revised paper to PLOS ONE. Your manuscript has now been assessed by our editorial team and previous peer experts, and I am pleased to inform you that your revised work has been approved by the reviewers. However, before I can recommend the final editorial decision to our journal office, some minor issues from Reviewer #4 need your attention. Please address them.

Reviewers' comments:

Reviewer's Responses to Questions

**Comments to the Author**

Reviewer #4: All comments have been addressed

2. Is the manuscript technically sound, and do the data support the conclusions?

Reviewer #4: Yes

3. Has the statistical analysis been performed appropriately and rigorously?

Reviewer #4: Yes

4. Have the authors made all data underlying the findings in their manuscript fully available?

Reviewer #4: Yes

5. Is the manuscript presented in an intelligible fashion and written in standard English?

Reviewer #4: Yes

Reviewer #4: Although it is good manuscript and author has addressed past comments. In some of the area still minor work needed. Such as in abstract numbers are provided (e.g., median scores, percentages, p-values), there’s no interpretive glue connecting them. For instance, the sentence: “Median social support and risk perception scores were 37 (63.79%) and 36 (40.0%) respectively” lacks context — is 40% alarmingly low? Is 63.79% moderate or high?

But overall this is acceptable.

**Do you want your identity to be public for this peer review?** For information about this choice, including consent withdrawal, please see our Privacy Policy

Reviewer #4: No

---

## [Author Response · Author response to Decision Letter 3]

7 Aug 2025

Thank you for your helpful feedback. In response to your concern that the abstract lacked “interpretive glue,” we have added explicit qualifiers to every numeric finding. Specifically:

1. The median social-support score is now labeled “moderate” (37 points; 63.79 % of the maximum).

2. The median risk-perception score is described as “alarmingly low” (36 points; 40.0 % of the maximum) and is further contextualized with severity and familiarity sub-scores rated as “very low” (32 %) and “low” (36.7 %), respectively.

3. All percentages, p-values, and correlation coefficients now include brief interpretive phrases so readers can immediately grasp their practical significance.

We have incorporated these changes into the revised abstract and uploaded the updated manuscript. Thank you again for guiding us toward a clearer presentation.

---

## [Editor Report · Decision Letter 3]

11 Aug 2025

Social Support and Risk Perception of Influenza Among Chengdu Residents: A Cross-Sectional Study During Post-Pandemic Recovery

PONE-D-25-07963R3

Dear Dr. Feng,

We’re pleased to inform you that your manuscript has been judged scientifically suitable for publication and will be formally accepted for publication once it meets all outstanding technical requirements.

Kind regards,

Xing-Xiong An, M.D.

Academic Editor

PLOS ONE

Additional Editor Comments (optional):

Thanks for the authors' efforts to comprehensively improve your manuscript according to editor's and reviewers' comments. I am pleased to inform you that your paper can be accepted for publication now.
---

## [Editor Report · Acceptance letter]

PONE-D-25-07963R3

PLOS ONE

Dear Dr. Feng,

I'm pleased to inform you that your manuscript has been deemed suitable for publication in PLOS ONE. Congratulations! Your manuscript is now being handed over to our production team.

Kind regards,

on behalf of

Dr. Xing-Xiong An

Academic Editor

PLOS ONE